# Effect by Diamond Surface Modification on Biomolecular Adhesion

**DOI:** 10.3390/ma12060865

**Published:** 2019-03-15

**Authors:** Yuan Tian, Karin Larsson

**Affiliations:** Department of Chemistry—Ångström Laboratory, Uppsala University, BOX 538, 75121 Uppsala, Sweden; yuan.tian@kemi.uu.se

**Keywords:** diamond, theory, biomolecules

## Abstract

Diamond, as material, show very attractive properties. They include superior electronic properties (when doped), chemical inertness, controllable surface termination, and biocompatibility. It is thus clear that surface termination is very important for those applications where the implant material is based on diamond. The present theoretical work has focused on the effect of diamond surface termination, in combination with type of surface plane, on the adhesion of important biomolecules for vascularization and bone regeneration. These biomolecules include Arginine-Glycine-Aspartic acid (RGD), Chitosan, Heparin, Bone Morphogenetic Protein 2 (BMP2), Angiopoietin 1 (AGP1), Fibronectin and Vascular Endothelial Growth Factor (VEGF). The various surface planes are diamond diamond (100)-2x1 and (111). The theoretical results show that the non-covalent binding of these biomolecules is in proportion with their molecular weights. Moreover, three groups of biomolecules were observed for both types of surface planes. The most strongly binding biomolecule was the BMP2 molecule. The smaller polypeptides (RGD, Chitosan and Heparin) formed a less strongly binding group. Finally, the biomolecules VEGF, Fibronectin and Angiopoietin showed bond strengths numerically in between the other two groups (thereby forming a third group). Moreover, the (111) surface was generally observed to display a stronger bonding of the biomolecules, as compared with the (100)-2x1 surface.

## 1. Introduction

Diamond is a material with very attractive properties. These include superior electronic properties (when doped), controllable surface termination, chemical inertness, high degree of biocompatibility, high transparency and a large electrochemical potential window, etc. When considering the well-known combination of chemical inertness, high transparency and high large electrochemical potential window [1,2], diamond became recently a promising candidate for applications like artificial photosynthetic water-splitting [3,4]. In addition, it has been observed that, because of the intrinsic biocompatibility and chemical stability, diamond can increase the duration of bone implants. Diamond will thereby become a preferred material for also biochemical applications [5,6]. 

Nano-crystalline diamond, or diamond material constructed from diamond nanoparticles with dimensions larger than ~3 nm in diameter, are today used for biochemical applications. The reactivity of these material surfaces is will depend on various factors: (i) diamond plane, (ii) surface termination and (iii) diamond particle size. It has both experimentally and theoretically been shown, that there is no size confinement for diamond particle diameters larger than ~2 nm [7,8]. In this work, it has therefore been assumed that nano-crystalline diamond or diamond particles of experimental sizes can be modelled using periodic supercells including the diamond low index planes (111) and (100)-2x1, respectively. The latter plane is a 2x1-reconstructed (100) surface.

A well-controlled surface termination with various species has not only proven to uphold the cubic structure of diamond, but also to change the reactivity and properties of the diamond surface region. For instance, specific surface termination can induce surface electronic conductivity and interfacial charge transfer properties. Other examples of properties that might be effected by surface termination are field emission characteristics [9], broad-band infrared reflectivity [10], hydrophilicity [11], electron conductivity [12] and electron affinity [13]. All these interesting properties make it clear that diamond surface termination is very important for especially those applications in which diamond can function as an electrode material [14].

Avascular necrosis causes cell death in various bone components. It is then common to replace the whole joints by artificial materials. However, the stability of these bone implant durations are not sufficiently high [15], so there is an urgent need to develop new biomaterials with increasing stability and biocompatibility. The biomaterials used today are hydroxyapatite, noble metals and polypropylenefumarate/polypropylenefumarate-diacrylate (PPF/PPF-DA). Hydroxyapatite may increase the biocompatibility, but does not show a good duration [16]. There are reports about resistant bacteria for noble metals, which have stopped its further use in the field [17]. Furthermore, local inflammations are a severe problem for the usage of PPF/PPF-DA implant materials [18]. However, diamond has proven to be an ideal biomaterial candidate for e.g., bone implants.

When developing new bone implant materials, a controllable interaction between growth factors and the material surface is of highest importance. Biocompatible diamond particles have recently become very interesting when studying these types of interactions [19]. Moreover, nanocrystalline diamond films, deposited onto titanium screws, have been proven to be osseoinductive [20,21]. However, there is at present a lack of scientific knowledge about diamond surfaces and their behavior in contact with biological molecules. One way to approach this problem is to use theoretical modeling as a highly valuable tool to study the interactions between diamond surfaces and adhered biomolecules. 

With the purpose to tailor-make the medical implant surfaces by utilizing the unique properties and versatility of diamond, a theoretical investigation about the interaction between diamond and biomolecules have in the present study been performed. The main goal with the present study has been to find the most optimal combination of crystallite facet and terminating species for which the respective biomolecule will bind strong enough and at the same time stay biological active. 

The main purpose with the present article is to theoretically outline the effect of diamond surface plane and termination on the binding of various important biomolecules. Diamond (111) and (100)-2x1 surface planes are then terminated with H, OH, O_ontop_, O_bridge_ and NH_2_ species, respectively. The various biomolecules include BMP2, RGD, chitosan, heparin, AGP1, fibronectin and VEGF [21]. The theoretical calculations are based on an ab initio force field method. 

## 2. Materials and Methods

It is, unfortunately, impractical to use first principle quantum mechanical method (like density functional theory (DFT) methods) for the present calculations. This is due to the large size of the models used in the present study (see Figure 1). However, these problems might be overcome by using force field (FF) methods. The advantage with force field methods is that they have the capacity to avoid the size restrictions of the DFT methods. However, the calculated bond energies (when using FF) has the tendency to become underestimated [22,23]. It is instead possible to use ab initio force fields that have been parameterized and validated using condensed-phase properties in addition to various ab initio and empirical data for molecules in isolation. This is a more accurate method to use for calculations of e.g., bond strengths in the systems of interests. The results in the present work have therefore been obtained by using the ab initio force field method within the program COMPASS (Condensed-phase Optimized Molecular Potentials for Atomistic Simulation Studies) from Accelrys, Inc. [24]. With this method it is possible to accurately predict structural properties for a broad range of biomolecules in isolation, as well as in condensed phases [25,26,27,28]. Since it is based on a force field method, it can be utilized for larger models up to 10^4^ atoms. 

The atomic charges used in the ab initio force field calculations where calculated by different means. The atomic charges for the diamond surface were calculated by using the DFT method (incorporated in CASTEP from Accelrys, Inc. [29,30,31]), whilst the atomic charges for the biomolecules were force field assigned in the FF calculations. The DFT method used in the present study is based on an ultrasoft pseudopotential plane-wave approach. The electronic exchange and correlation have been approximated using the Generalized Gradient Approximation as developed by Perdew, Burke and Ernzerhof (GGA-PBE) [32]. This specific approximation is supposed to be more accurate for the calculations of surface related properties [33]. Furthermore, the assigned atomic charges are Mulliken charges. Hirsfield charges would be an alternative, but that type of atomic charge often underestimates the charges distribution in molecules due to its dependence mainly on free-atom electron densities [34]. According to the Mulliken method for calculating atomic charges, a projection of the plane wave states onto the localized basis set was used in the present study [35].

For the ab initio FF optimization calculations, the threshold values were set to 8.4 × 10^−5^ kJ/mol (energy), 0.042 kJ/mol/Å (force) and 1.0 × 10^−5^ Å (displacement), respectively. An ordinary geometry optimization procedure initiated each simulation, which was followed by the real annealing process. The purpose this annealing process is to simulate the experimental heat treatment process in finally reaching the static state of the system under investigation. It is widely used in predicting protein adhesion affinities [26,36]. The starting temperatures in the annealing processes in the present study were set to 300 K and the mid-cycle temperature was set to 320 K. This is the temperature range that has been found appropriate to use for the physiological functionality of biomolecules. The duration of the simulations was ~25 ps and ended by an ordinary geometry optimization. The lengths of these simulations were rigorously tested and where then found long enough to reach energetic minima. 

All atoms in the models, except the two bottom atomic layers, were allowed to move freely during the simulations. The two atomic bottom layers had to be kept fixed in order to simulate the continuation into a bulk structure. This type of constraint has earlier been shown to be valid for diamond systems [37]. 

The adhesion energies for the biomolecules, attached to the various diamond surfaces, were calculated by using Equation (1):ΔE_adhesion energy_ = (E_diamond-biomolecule_ − E_surface_ − E_biomolecule_)(1)
where E_diamond-biomolecule_ is the energy for the whole system, E_surface_ is the energy for the diamond surface and E_biomolecule_ is the energy for the biomolecule, respectively. The final structural geometries, obtained from a geometry optimization of the whole systems, where used for these single point energy calculations.

The construction and use of a realistic model are very important when performing theoretical investigations. A number of criteria must then be fulfilled. Earlier theoretical studies have shown that the models used for the diamond (100)-2x1 and (111) surfaces in the present study, is optimal to use for the present type of investigation [38]. The supercell dimensions for the diamond (100)-2x1 and (111) surfaces are: a = 80.46 Å, b = 90.52 Å and c = 85.11 Å and: a = 80.46 Å, b = 90.52 Å and c = 80.00 Å, respectively.

The lower part of the periodic surface slab is H-terminated. The reason for this termination is to simulate the continuation into bulk diamond. Moreover, the upper surface is either H–, O_bridge_–, O_ontop_–, OH–, or NH_2_–terminated (see Figure 1B,C). An earlier theoretical investigation showed that it is energetically possible (and even favorable) to terminate a diamond (100)-2x1 surface up to 100% with either H, O or OH species [39]. A 100% surface coverage by these species was therefore been used in the present study. It has also been found possible to experimentally and precisely, oxidize the diamond surfaces up to larger surface coverages [40].

The various biomolecules that have been used in the present study are: RGD, chitosan and heparin, as well as BMP2, AGP1, fibronectin and VEGF. The latter group of biomolecules are considerably smaller than the former ones (which are quite larger proteins). However, all of these biomolecules are growth factors which promote new blood vessel formation, cell growth and adhesion. The models of these biomolecules were taken from the Protein Data Bank [41]. These respective biomolecules were initially attached to the diamond surface with the protein-binding pocket positioned upwards with respect to the diamond surface (i.e., where ligands can easily bind to it). The purpose with this type of relative position was to keep the functionality of the biomolecule. A geometry-optimized structure of this type of system can be seen in Figure 1A. More specifically, it is a model of an H-terminated diamond (111) surface with an attached VEGF biomolecule. 

## 3. Results

### 3.1. Introduction

As presented in Section 2, the models of the diamond//biomolecule systems were too big to use for the more accurate DFT methods. The methods that though can be used for these systems are the ab initio force field methods. In order to validate the usefulness of these methods, test calculations were performed in a recent work by the present authors [42]. Results from the more accurate ab initio FF method were then compared with the results obtained by using a hybrid DFT/FF method. The calculated adhesion energies for large physisorbed organic molecules onto differently terminated diamond surface planes thereby showed that the ab initio FF method is possible to use in the present study. 

### 3.2. Adhesion Energies for Biomolecules Attached to Diamond (100)-2x1 and (111) Surfaces

The combined effect of surface termination and type of diamond surface plane has in the present study been theoretically investigated in order to investigate the adhesion strength for various biomolecules within the field of bone regeneration and vascularization. The diamond surface planes studied were the (111) surface and the 2x1-reconstructed (100) surface. Moreover, the terminating species include H, OH, O_ontop_, O_bridge_ and NH_2_. 

#### 3.2.1. Effect of a Terminated Diamond (100)-2x1 Surface on Biomolecule Adsorption

For the (100)-2x1 surface (see Table 1), the adhesion energy of BMP2 was observed to be numerically higher than any of the other candidates. The adhesion energy was more than 2–3 times larger than the other biomolecules. This was the situation for the O_ontop_–, O_bridge_–, OH– and NH_2_–terminating species, with a special emphasize on the O_ontop_ and O_bridge_ species. On the contrary, the results for the H–terminating species was very similar to corresponding results for some of the other biomolecules; AGP, VEGF and fibronectin [2312 (BMP2), 1460 (AGP1), 1580 (VEGF) and 2090 (fibronectin) kJ/mol].

Van der Waals interactions between e.g., a biomolecule and a surface depend on the molecular weight of the biomolecule. Since the number of atoms within BMP2 (3860) is within the same order of magnitude as the other three proteins (AGP1 3383 atoms; Fibronectin 2944 atoms; VEGF 1531 atom), their respective molecular weight must also be similar (see Table 2). Hence, the difference in adsorption strengths for the various biomolecules, is not only due to differences in dispersive forces (i.e., Van der Waals interactions). There must also be a contribution of electrostatic interactions and/or hydrogen bonding to the extraordinary bond strength situation of BMP2. This assumption is strongly supported by the fact that BMP2 reacts especially strongly with the highly polar O_ontop_- and O_bridge_-terminated diamond surfaces. A strong binding of BMP2 onto a diamond surface has also been observed experimentally. This is an experimental result that strongly supports our theoretical observations. The bond strength between an attached BMP2 and the diamond surface were thereby measured by XPS-analysis (X-ray photoelectron spectroscopy) and the binding affinities were found to be much higher compared to the situation when using pure titanium as the solid surface. In addition, the bioactivity of BMP2 was retained as a result of the binding to the surface when using a diamond surface [43,44].

The second group of biomolecules, as shown in Table 1, is composed of VEGF, fibronectin and AGP1. The sizes of these biomolecules are around several thousands of atoms. Their adhesion energies are therefore quite large (750–2090 kJ/mol) and they thereby form a middle group (energetically) of biomolecules within the present study. For this middle group, H-termination was, with one exception, shown to result in the strongest adhesion energy to the diamond (100)-2x1 surface. The exception was the VEGF biomolecule, which showed the highest adhesion energy for NH_2_-termination. Since the molecular weight is most important for the Van der Waals-interaction between a diamond surface and the adhered biomolecule interaction and since VEGF is only half the size of angiopoietin 1, it would be expected that the adhesion energy of AGP1 is larger than for the VEGF biomoleule. However, the adhesion energies are, with the exception of O_bridge_-termination, very similar for these two biomolecules.

The third biomolecular group includes the much smaller polypeptides: RGD, chitosan and heparin. Their adhesion energies (10–800) were also found to be smaller in comparison to the other biomolecules, which is again a proof of the circumstance that the sizes of the biomolecules are of utter importance for size of the Van der Waals interactions. Furthermore, the H-termination was shown to give the largest adhesion energy for all of these polypeptides.

#### 3.2.2. Effect of a Terminated Diamond (111) Surface on Biomolecule Adsorption

The adhesion energies for the different biomolecules adsorbed onto the diamond (111) surface, are shown in Table 3. It has earlier been shown that the diamond (111) surface is more reactive than the (100)-2x1 surface [45]. It is the reactivity of the individual surface carbon atoms that is referred to. Moreover, the densities of surface C atoms on the (111) and (100)-2x1 surfaces, are 0.19 and 0.16 atoms/Å^2^, respectively. The combination of a larger chemical reactivity of individual surface C atoms and a larger surface atom density for the (111) surface, is thereby expected to result in larger adhesion energies for the biomolecules. This was especially the situation with the largest biomolecules that covered a larger part of the diamond surfaces (BMP2, VEGF, fibronectin, AGP1); 1150–8200 kJ/mol for the (111) surface and 750–5480 kJ/mol for the (100)-2x1 surface.

As was the situation for the diamond (100)-2x1 surface, three adhesion energy groupings of biomolecules were found for the diamond (111) surface (also here displaying the importance of molecular weight for the surface-biomolecule Van der Waals interactions). As can be seen in Table 3, it is obvious that O_bridge_–termination is the preferred type of termination. This is in fact the situation for BMP2 (which from here on is called group 1) and for all molecules in group 2. Moreover, H–termination was, for group 1, found to give the weakest adhesion energy. Also, fibronectin VEGF and (in group 2) were shown to have the weakest adhesion energies for an H-terminated diamond (111). On the contrary, AGP1 showed the weakest adhesion for O_ontop_–termination. In the third and most weakly bound group of biomolecule adsorbates, heparin and chitosan were shown to both prefer O_bridge_–termination and disfavor H-termination. In addition, H–termination was energetically favored for RGD. The lowest adhesion energy in the present study was observed for the RGD peptide onto an O_ontop_–terminated (111) surface, with a numerical value as low as 12 kJ/mol.

### 3.3. Adhesion Energies for Various Types of Surface Termination

For H–terminated diamond surfaces, the calculated adhesion energies for adsorbed BMP2, AGP1 and VEGF biomolecules were found to be larger for the diamond (111) surface than for diamond (100)-2x1 (see Figure 2). The situation is, however, the opposite for RGD and Fibronectin. As stated above, the adhesion energies are usually more pronounced for diamond (111) surfaces due to the combination of both its higher surface carbon density and the large chemical reactivity of each individual surface C atom (i.e., the (111) surface has a larger surface energy as compared to (100)-2x1). However, an exception from this rule is here evident for RGD and Fibronectin and the explanation may be an existence of electrostatic interactions between the H–terminated surface and these biomolecules. It should also be stressed that the difference in calculated adhesion energy, with respect to the other biomolecules, is relatively small for both of these biomolecules. For OH–termination, the calculated adhesion energies for all biomolecules attached to diamond (111) were found to be larger than for those attached to diamond (100)-2x1. Moreover, the results for the O_ontop_–termination showed the same major trend as for the OH–termination.

As was the situation with the OH–, O_ontop_– and O_bridge_–terminations, the adhesion energies for all biomolecules attached to an NH_2_–terminated diamond (111) surface were observed to be numerically similar to, or larger than, the corresponding adhesion energies for diamond (100)-2x1. It should also be noted that the adhesion energies for both the diamond (100)-2x1 and (111) surfaces, were observed to be not that large (i.e., almost same) for the smallest biomolecular groups.

### 3.4. Adhesion Energies for Individual Biomolecules

As presented in Section 3.2. Adhesion energies for biomolecules attached to diamond (100)-2x1 and (111) surfaces, the adhesion energies for the biomolecules BMP2, VEGF, AGP1, fibronectin, heparin, chitosan and RGD have generally been found to be larger for the diamond (111) surface than for diamond (100)-2x1 (see Figure 3). This is especially the situation for the larger biomolecules (i.e., for fibronectin, VEGF, AGP1 and BMP2. As discussed above, this may be explained by differences in surface energy for the two surface planes. Two exceptions to this general observation are the adhesion of RGD and fibronectin to H-terminated diamond surfaces. As can be seen in Figure 2a, it is the adsorption of the various biomolecules to an H-terminated (100)-2x1 surface that results in a somewhat more irregular shape of the adsorption energy curve. The reason to this observation might be changes in degree of surface reconstruction. A de-reconstructed diamond (100) surface is more reactive than a 2x1-reconstructed (100) surface. In addition, BMP2 shows much larger adhesion energies as compared to the rest of the growth factors, which correspond to earlier experimental findings [44]. Specifically, for the BMP2 adhesion onto the diamond (111) surface, the order of adhesion energy for the various surface-terminating species was calculated as O_bridge_ > OH > NH_2_ > O_ontop_ > H. However, the corresponding order for the diamond (100)-2x1 surface was observed to be different; O_ontop_ > O_bridge_ > NH_2_ > OH > H. More specifically, it is the OH– and O_ontop_–termination that will break an otherwise perfect similarity in trend between these two diamond surfaces. For the situation with VEGF, the order of preference for the various surface termination species onto the diamond (111) surface was found to be O_bridge_ > NH_2_ > O_ontop_> OH > H. The corresponding order for the diamond (100)-2x1 surface is NH_2_ > O_ontop_ > H > O_bridge_ > OH. For this biomolecule, in addition to OH-termination and oxygen in bridge positions (i.e., O_bridge_–termination) that will break an otherwise perfect similarity in trend between these two diamond surfaces.

The trends in adhesion energy as a function of surface termination type, were also observed to be very similar for the AGP1 protein: O_bridge_ > H > NH_2_ > OH > O_ontop_ (diamond (111)) and O_bridge_ > H > NH_2_ > O_ontop_ > OH (diamond (100)-2x1). It is here only the O_ontop_–termination that will break an otherwise perfect similarity in trend between these two diamond surfaces.

The corresponding trends in adhesion energies were observed to be completely different for fibronectin. The order of adhesion energies for the diamond (111) surface was calculated as O_bridge_ > NH_2_ > OH > O_ontop_ > H. Moreover, for the diamond (100)-2x1 surface it was H > NH_2_ > O_bridge_ > O_ontop_ > OH. Experimental studies using AFM (atomic force microscopy) and FTIR (Fourier-transform infrared) spectroscopy have indicated that fibronectin can absorb onto Si surfaces. The surface chemical properties (especially the wettability) were observed to play an important role in keeping the structure and functionality of the protein. For a hydrophilic Si surface, the alpha-helix content in adsorbed fibronectin was observed to be similar to the original (molecular) structure with a good consistency [46].

For heparin, the order of preferences for adhesion onto the (111) surface was observed to be O_bridge_ > NH_2_ > OH > O_ontop_ > H and for diamond (100)-2x1 surface it was O_ontop_ > H > NH_2_ > O_bridge_ > OH. As was the situation for fibronectin, these two different trends are completely dissimilar. However, the orders of adhesion energies, when comparing the adhesion of fibronectin and heparin onto diamond (111), were found to be identical. For the situation with diamond (100)-2x1, the orders were identical, with the exception for O_ontop_–termination.

For chitosan, the calculated adhesion energies for the diamond (111) and (100)-2x1 surfaces were; O_bridge_ > NH_2_ > OH > O_ontop_ > H and H > O_ontop_ > NH_2_ > O_bridge_ > OH, respectively. As was the situation for heparin and fibronectin, these two different trends are completely dissimilar. Moreover, the orders of adhesion energies, when comparing the adhesion of fibronectin, heparin and chitosan onto diamond (111), were found to be identical. For the situation with diamond (100)-2x1, the orders were also here identical with the exception for O_ontop_–termination.

For the RGD peptide, the order of preferences for adhesion to variously terminated diamond (111) surfaces were calculated to be H > O_bridge_ > OH > NH_2_ > O_ontop_ and for terminated diamond (100)-2x1 surfaces it became H > O_ontop_ > O_bridge_ > OH > NH_2_. For this biomolecule, it is the H-termination (diamond (111)) and H– and OH–termination (diamond (100)-2x1), that will break an otherwise perfect similarity in trend between these two diamond surfaces.

## 4. Discussion

With the purpose to tailor-make the bone implant surfaces by using the unique surface properties of diamond, a theoretical investigation has in the present study been performed in order to more deeply study the interaction between diamond and various biomolecules. More specifically, the main goal has been to find the most optimal combination of diamond surface plane and terminating species for which various biomolecules will bind strong enough and at the same time stay biological active.

The H, OH, O_ontop_, O_bridge_ and NH_2_ species were in the present study used to completely terminate the diamond (111) and (100)-2x1 surface, respectively. The biomolecules that were attached to these terminated surfaces were BMP2, RGD, chitosan, heparin, AGP1, fibronectin or VEGF. The calculated results were obtained by using the ab initio force field methods within the COMPASS software from Accelrys, Inc.

The results show that the sizes of the individual biomolecules are very important for the adhesion energies. Three groups of biomolecules were observed for both the diamond (100)-2x1 and (111) planes. The largest biomolecular, BMP2, resulted in the strongest binding. On the contrary, the weakest binding was obtained for the smaller polypeptides: RGD, chitosan and heparin. A third group, with adhesion energies somewhere in between the other two groups, contained VEGF, fibronectin and AGP1.

The combination of diamond surface plane and termination type resulted in predominant variations in adhesion energies for the various biomolecules. In relation to diamond (100)-2x1, OH–, O_ontop_–, O_bridge_– and NH_2_–terminated diamond (111) surfaces resulted in stronger bonding with the biomolecules. The situation was different for H-terminated surfaces. The biomolecules RGD and fibronectin were observed to bind stronger onto the H–terminated diamond (100)-2x1 surface. An overview of the combined diamond plane—surface termination impact on the adhesion energy, for various biomolecules, is shown below.

For the BMP2 biomolecule, the order of adhesion energy for the diamond (111) surface was: O_bridge_ > OH > NH_2_ > O_ontop_ > H and for the diamond (100)-2x1 surface it was: O_ontop_ > O_bridge_ > NH_2_ > OH > H. For the VEGF biomolecule, the order of adhesion energies onto the diamond (111) surface was: O_bridge_ > NH_2_ > O_ontop_> OH> H and for the diamond (100)-2x1 surface it was: NH_2_ > O_ontop_ > H >O_bridge_ > OH. For the AGP1 biomolecule, the trends in adhesion energy for both the diamond (100)-2x1 and (111) surfaces were found to be identical; O_bridge_ > H > NH_2_ > OH > O_ontop_. For the fibronectin biomolecule, the order of adhesion energies onto the diamond (111) surface was: O_bridge_ > NH_2_ > OH > O_ontop_ > H. The corresponding order for the diamond (100)-2x1 surface was: H > NH_2_ > O_bridge_ > O_ontop_ > OH. For the heparin biomolecule, the order of adhesion energies onto the diamond (111) surface was: O_bridge_ > NH_2_ > OH > O_ontop_ > H and for the diamond (100)-2x1 surface it was: O_ontop_ > H > NH_2_ > O_bridge_ > OH. For the chitosan biomolecule, the order of adsorption energies for diamond (111) was: O_bridge_ > NH_2_ > OH > O_ontop_ > H and for diamond (100)-2x1 it was: H > O_ontop_ > NH_2_ > O_bridge_ > OH. The weakest adhesion energies in the present study was obtained for the RGD peptide, with the following result for the diamond (111) surface: H > O_bridge_ > OH > NH_2_ > O_ontop_. This is to be compared with the results for diamond (100)-2x1: H > O_ontop_ > O_bridge_ > OH > NH_2_.

There are many factors that might influence the adhesion energy for biomolecules attached to a diamond surface. The most important ones are (i) size of biomolecule, (ii) surface plane, (iii) surface termination and (iv) surface reconstruction. The influences of these parameters have been discussed above in close connection to the obtained results in the present study. General trends were thereby found for each of the different biomolecules. However, other factors might influence the obtained results, thereby introducing exceptions to the general trends. For instance, the interactions between the diamond surface and the biomolecules may cause changes in the surface reconstructions (i.e., the 2x1-reconstructed (100) surface may de-reconstruct to a (100) surface). Moreover, the initial terminating species may undergo changes and thereby form another type of termination (i.e., the very reactive O_bridge_–terminated diamond (111) surface may become O_ontop_–terminated). It is very difficult to explain all individual results in the present study, but plausible explanations can be given. For instance, the larger adhesion energies for the larger biomolecules being attached to an initial O_bridge_–terminated diamond (111) surface might be due to the transformation to the more stable O_ontop_–terminated diamond (111) surface. Moreover, the reason to the somewhat larger adhesion energies for the OH– and NH_2_–terminated diamond (111) surface might be the formation of a larger extent of H bonds between the biomolecule and the surface. Another explanation could be interfacial reactions with the occurrence of an H transfer from the surface to the respective biomolecule. Due to the size of the model systems used in the present investigation, it is practically not possible to analyze these influencing factors more in detail.

The main result from the following study is that H-termination is generally good to use for the diamond (100)-2x1 surface, whilst O_bridge_–termination is a better choice for the diamond (111) surface. In addition, BMP2 shows the absolutely highest adhesion energies of all candidates in the present study.

## Figures and Tables

**Figure 1 materials-12-00865-f001:**
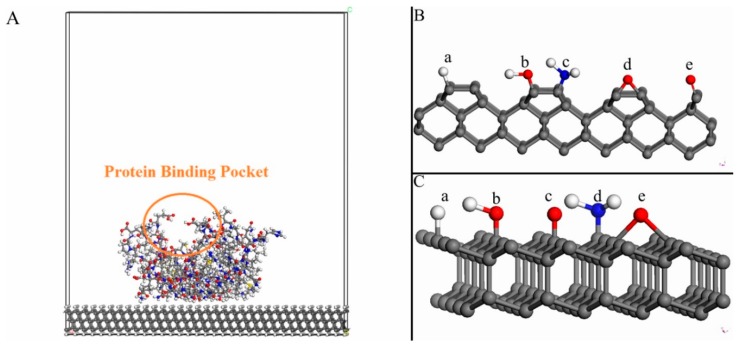
(**A**) Model demonstrating a 2x1-reconstructed diamond (111) surface that is to 100% terminated with H atoms and thereafter attached to a Vascular Endothelial Growth Factor (VEGF) molecule. The orientation of the biomolecule is such that the protein binding pocket is on top and far away from the surface. (**B**) Five different termination species adsorbed onto a 2x1-reconstructed diamond (100) surface: (a) H, (b) OH, (c) NH_2_, (d) O_bridge_ and (e) O_ontop_. (**C**) Five different termination species that are adsorbed to a diamond (111) surface: (a) H, (b) OH, (c) O_ontop_, (d) NH_2_ and (e) O_bridge_.

**Figure 2 materials-12-00865-f002:**
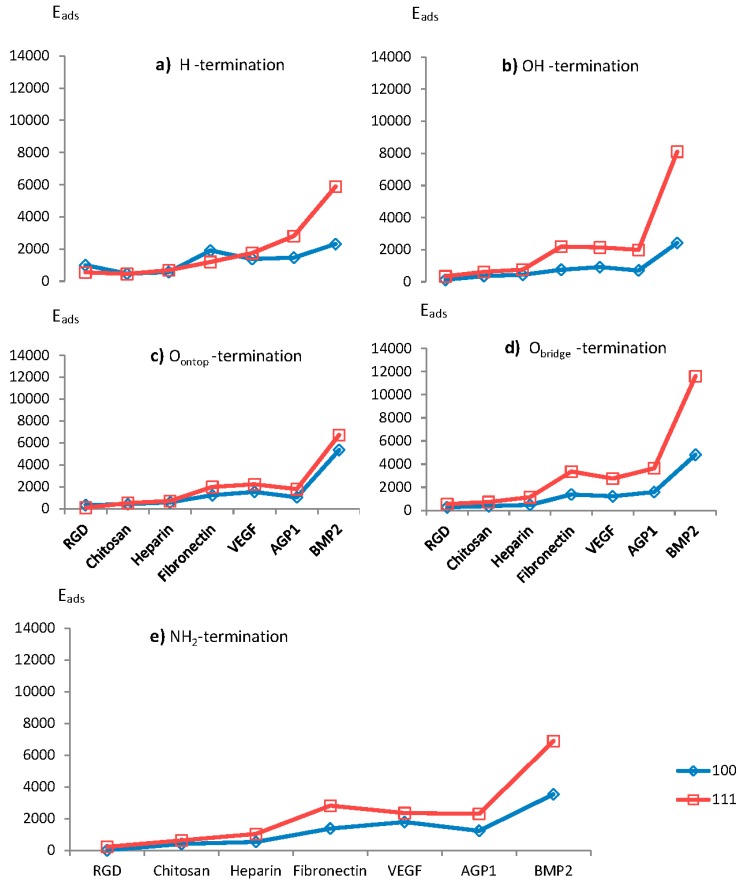
Adhesion energies (in kJ/mol) for various growth factors attached to diamond (100) and (111) surfaces, being terminated with (**a**) H, (**b**) OH, (**c**) O_ontop_, (**d**) O_bridge_, or (**e**) NH_2_ species.

**Figure 3 materials-12-00865-f003:**
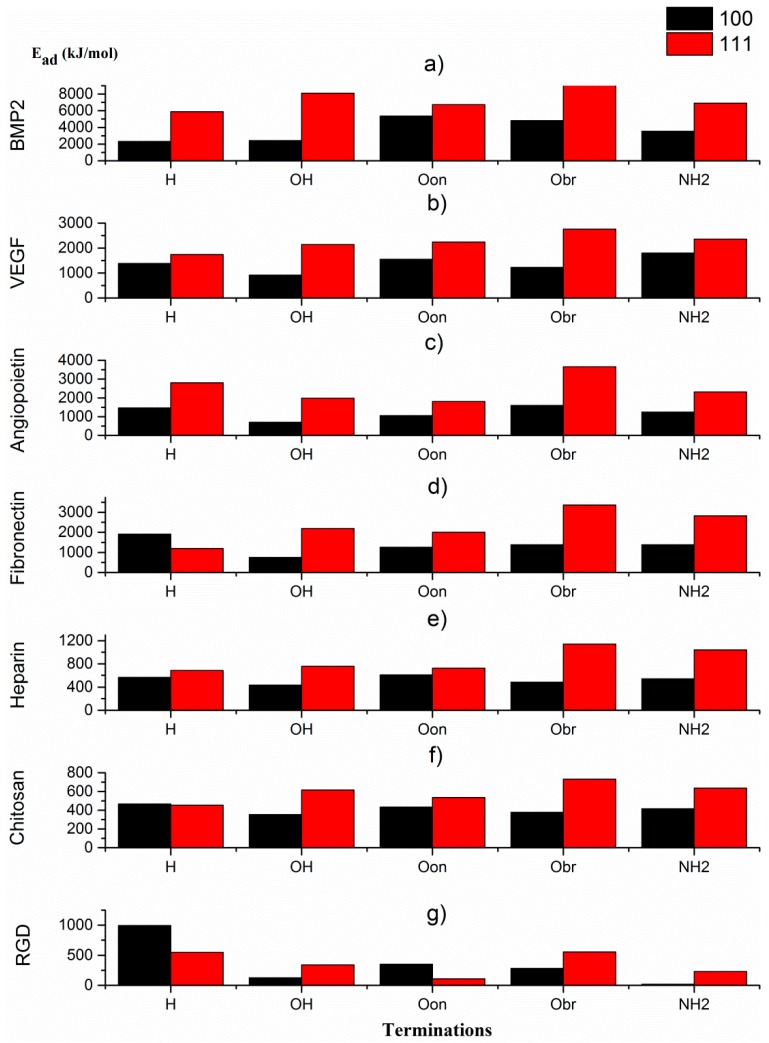
Adhesion energies as a function of surface termination species for seven different biomolecules attached to diamond (111) and (100)-2x1, respectively; (**a**) BMP2, (**b**) VEGF, (**c**) AGP1, (**d**) fibronectin, (**e**) heparin, (**f**) chitosan and (**g**) RGD peptide. The surface terminating species are H, OH, O_ontop_, O_bridge_ and NH_2_.

**Table 1 materials-12-00865-t001:** Adhesion energy for various biomolecules attached onto an X-terminated diamond (100)-2x1 surface (X = H, OH, O_ontop_, O_bridge_ and NH_2_). The biomolecules include VEGF, BMP2, AGP1, fibronectin, heparin, chitosan and RGD. The unit is kJ/mol.

Diamond (100)-2x1	H	OH	O_ontop_	O_bridge_	NH_2_
BMP2	2312	2480	5480	5202	3803
VEGF	1580	970	1640	1140	1840
Fibronectin	2090	805	1204	1450	1195
AGP1	1460	750	838	1448	1115
Heparin	601	495	520	512	535
Chitosan	520	480	525	485	535
RGD	800	30	470	460	10

**Table 2 materials-12-00865-t002:** Number of atoms for the biomolecules BMP2, VEGF, fibronectin, AGP1, heparin, chitosan and RGD.

Biomolecules	Number of Atoms in the Molecular Unit
BMP2	3860
VEGF	1531
Fibronectin	2944
AGP1	3383
Heparin	112
Chitosan	16
RGD	46

**Table 3 materials-12-00865-t003:** Adhesion energy for various biomolecules attached onto an X-terminated diamond (111) surface (X = H, OH, O_ontop_, O_bridge_ and NH_2_). The biomolecules include VEGF, BMP2, AGP1, fibronectin, heparin, chitosan and RGD. The unit is kJ/mol.

Diamond (111)	H	OH	O_ontop_	O_bridge_	NH_2_
BMP2	5800	8007	6420	11,940	6428
VEGF	1923	2040	2043	2302	2172
Fibronectin	1103	2038	2003	2803	2401
AGP	2940	1980	1966	3543	2008
Heparin	641	639	635	704	680
Chitosan	502	605	603	651	602
RGD	641	248	12	458	58

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
