# Peer review of "Effect by Diamond Surface Modification on Biomolecular Adhesion"

_materials, 2019, doi:10.3390/ma12060865_

Reviewer 1 Report

The study deals with possible applications of diamond in medicine. The results of the study can be important for the development of implants technology using diamond.

There is a minor suggestion to elucidate better a state of the art diamond technology for implants preparation.

1.     In introduction, it is mentioned “diamond-covered implants“. The study investigates only single crystal diamond (sp3 structure) substrates. The growth of large size single crystal is still significant technological problem, as well as probably only small area single crystal diamond can be used as implant. Nowadays, probably only MCD and NCD films (sp2 and sp3 structures) can be prepared on the large size implants. What authors think about the considering adhesion not only between sp3 and molecules but more realistic between sp2+sp3 types of surface and molecules? Probably, for the future studies these type of calculation would be very desirable?

2.     On the graph in Fig. 2 should be Ead(kJ/mol)?

Author Response

Reviewer #1:

Comment 1:    

In introduction, it is mentioned “diamond-covered implants“. The study investigates only single crystal diamond (sp3 structure) substrates. The growth of large size single crystal is still significant technological problem, as well as probably only small area single crystal diamond can be used as implant. Nowadays, probably only MCD and NCD films (sp2 and sp3 structures) can be prepared on the large size implants. What authors think about the considering adhesion not only between sp3 and molecules but more realistic between sp2+sp3 types of surface and molecules? Probably, for the future studies these type of calculation would be very desirable?

Answer 1:

In the Introduction of the present manuscript, we discuss the choice of model in the present study.  We there present the circumstance that diamond nanoparticles with dimensions larger than ~3 nm in diameter are nowadays used for biochemical applications. Moreover, it has both experimentally and theoretically been shown, that there is no size dependency found for diamond particle diameters larger than ~2 nm. It is therefore possible to use super cell diamond planes in modelling the corresponding planes on these types of larger diamond nanoparticles. As mentioned by the Reviewer, there are also other C sites on the nanoparticles (e.g., sp2 C). However, the occurrences of these C sites are minor in comparison to the C sites on the diamond planes. Despite of this, it would be very interesting to also investigate the influence of these sites in a future investigation.

Reviewer 2 Report

1.) A table with some parameters (e.g. number of atoms, molecule mass) of the modeled biomolecules will be useful for the readers.

2.) Please clarify “(100)-2x1 surface”

3.) 3.1 section details results of another paper of the same authors. I suggest to shorten this section and instead of citing the details underlie the usefulness of ab initio FF method in front of ordinary DTF method.

4.) I don’t see justified to define a group with a single member when discussing the results of the calculations.

5.) Significance of differences between adhesion energies should be evaluated. E.g. in lines 280-283 is written that for chitosan (100)-H 520kJ/mol and (111)-H 502kJ/mol difference<5%. The case of fibronectin is evaluated as similar, however there is a factor of two between the adhesion energies of (100)-H (2090 kJ/mol) and (111)-H (1103kJ/mol) surfaces.

Similarly for AGP1 (111)-OH and (111)-O ontop, lines 297-299.

For the same reasons I do not agree with the general recommendation in lines 398-399, which should be omitted (vs. VEGF, AGP1 and chitosan data in Table 1)

6.) Discussion should offer at least some tentative explanation for the findings. E.g. why generally OH terminations have the lowest adhesion energy on (100) surface and Obridge the highest on (111) surface.

Or why RDG has the highest adhesion energy on H terminated (111) surface while all other modeled molecules prefer O bridge on (111) surface.

What can be the reason for the order of magnitude difference between RDG (111)-H and (111)-O ontop. Similarly, RDG (100)-H and (100)-OH, NH2

7.) Units should be given in Figure 2.

8.) The (100) surface is favoured in two cases. This should be identified, outlined and explained.

9.) The first paragraph of the 4. Discussion section is practically the repetition of the Introduction. It is unnecessary, thus should be deleted.

Author Response

Reviewer #2:

Comment 1:

A table with some parameters (e.g. number of atoms, molecule mass) of the modeled biomolecules will be useful for the readers.

Answer 1:

 3.1 section details results of another paper of the same authors. I suggest to shorten this section Answer 1:

A table with number of atoms for the various biomolecules are included in the text. (We didn´t include the molecule mass since we didn´t find that information in the literature). The numbers of the rest of the Tables are also changed.

------------------------------------------------------------------------------------------------

Comment 2:  

Please clarify “(100)-2x1 surface”

Answer 2:

We have included the sentence “The latter plane is a 2x1-reconstructed (100) surface.” in section 1. Introduction.

------------------------------------------------------------------------------------------------

Comment 3:

and instead of citing the details underlie the usefulness of ab initio FF method in front of ordinary DTF method.

Answer 3:

We have shortened the section as suggested by the authors.

------------------------------------------------------------------------------------------------

Comment 4:

I don’t see justified to define a group with a single member when discussing the results of the calculations.

Answer 4:

We have in section 3.2.2. Effect of surface-termination on biomolecule adsorption for diamond (111) surfaces declared that group 1 only include one biomolecule.

------------------------------------------------------------------------------------------------

Comment 5:

Significance of differences between adhesion energies should be evaluated. E.g. in lines 280-283 is written that for chitosan (100)-H 520kJ/mol and (111)-H 502kJ/mol difference<5%. The case of fibronectin is evaluated as similar, however there is a factor of two between the adhesion energies of (100)-H (2090 kJ/mol) and (111)-H (1103kJ/mol) surfaces.

Similarly for AGP1 (111)-OH and (111)-O ontop, lines 297-299.

For the same reasons I do not agree with the general recommendation in lines 398-399, which should be omitted (vs. VEGF, AGP1 and chitosan data in Table 1)

Answer 5:

We made a mistake in the text and made necessary corrections (see sections 3.4. Adhesion energies for individual biomolecules and 4. Discussion.

------------------------------------------------------------------------------------------------

Comment 6:

Discussion should offer at least some tentative explanation for the findings. E.g. why generally OH terminations have the lowest adhesion energy on (100) surface and Obridge the highest on (111) surface.

Or why RDG has the highest adhesion energy on H terminated (111) surface while all other modeled molecules prefer O bridge on (111) surface.

What can be the reason for the order of magnitude difference between RDG (111)-H and (111)-O ontop. Similarly, RDG (100)-H and (100)-OH, NH2

Answer 6:

We have added plausible explanations for the findings in section 4. Discussion.

----------------------------------------------------------------------------------------------------------

Comment 7:

Units should be given in Figure 2.

Answer 7:

We have changed “Ead” to “Eads” in Figure 2 and included “(in kJ/mol)” in the figure text for Figure 2.

----------------------------------------------------------------------------------------------------------

Comment 8:

The (100) surface is favoured in two cases. This should be identified, outlined and explained.

Answer 8:

We have added an explanation to this observation in section 4. Discussion.

---------------------------------------------------------------------------------------------------------------

Comment 9:

The first paragraph of the 4. Discussion section is practically the repetition of the Introduction. It is unnecessary, thus should be deleted.

Answer 9:

We have omitted this paragraph.

------------------------------------------------------------------------------------------------

Reviewer 3 Report

The paper deals with ab initio simulations of the different adsorption energy biomolecules and peptides depending on the linking functional group onto surface diamond. I find the manuscript really interesting and helpful as shed lights about many phenomenological experimental results and clarify as well as promotes a proper selection and optimization of functionalization strategies in order to obtain the best adsorption and conformation possible. Although, more studies are needed, I recommend the study for publication.

Minor concerns:

Correct typos such as “hydroxiapatit”.

Author Response

Reviewer #3:

The paper deals with ab initio simulations of the different adsorption energy biomolecules and peptides depending on the linking functional group onto surface diamond. I find the manuscript really interesting and helpful as shed lights about many phenomenological experimental results and clarify as well as promotes a proper selection and optimization of functionalization strategies in order to obtain the best adsorption and conformation possible. Although, more studies are needed, I recommend the study for publication.

Minor concerns:

Correct typos such as “hydroxiapatit”.

Answer 1:

I have corrected “hydroxiapatit” to “hydroxyapatite”.

Round  2

Reviewer 2 Report

Comment1:

A table with some parameters (e.g. number of atoms, molecule mass) of the modeled biomolecules will be useful for the readers.

Answer 1:

 3.1 section details results of another paper of the same authors. I suggest to shorten this section Answer 1:

A table with number of atoms for the various biomolecules are included in the text. (We didn´t include the molecule mass since we didn´t find that information in the literature). The numbers of the rest of the Tables are also changed.

Comment on answer 1: OK.

------------------------------------------------------------------------------------------------

Comment2:  

Please clarify “(100)-2x1 surface”

Answer 2:

We have included the sentence “The latter plane is a 2x1-reconstructed (100) surface.” in section 1. Introduction.

Comment on answer 2: OK

------------------------------------------------------------------------------------------------

Comment3:

and instead of citing the details underlie the usefulness of ab initio FF method in front of ordinary DTF method.

Answer 3:

We have shortened the section as suggested by the authors.

Comment on answer 3: OK

------------------------------------------------------------------------------------------------

Comment4:

I don’t see justified to define a group with a single member when discussing the results of the calculations.

Answer 4:

We have in section 3.2.2. Effect of surface-termination on biomolecule adsorption for diamond (111) surfaces declared that group 1 only include one biomolecule.

Comment on answer 4:

I still do not see justified group 1, and my opinion is supported by the new Table 2 which contains  the number of atoms in each biomolecule considered in the calculations. Considering this parameter (i.e. the number of atoms in the biomolecule) AGP 1 is much more similar to BMP2 than to VEGF, however, it has been put in the same group with VEGF by the authors.

If the authors find it justified to define a separate group for BMP2, they should give the reason for it, i.e. present a (set of) parameters that support the extreme high adhesion energy of BMP2.

------------------------------------------------------------------------------------------------

Comment5:

Significance of differences between adhesion energies should be evaluated. E.g. in lines 280-283 is written that for chitosan (100)-H 520kJ/mol and (111)-H 502kJ/mol difference<5%. The case of fibronectin is evaluated as similar, however there is a factor of two between the adhesion energies of (100)-H (2090 kJ/mol) and (111)-H (1103kJ/mol) surfaces.

Similarly for AGP1 (111)-OH and (111)-O ontop, lines 297-299.

For the same reasons I do not agree with the general recommendation in lines 398-399, which should be omitted (vs. VEGF, AGP1 and chitosan data in Table 1)

Answer 5:

We made a mistake in the text and made necessary corrections (see sections 3.4. Adhesion energies for individual biomolecules and4. Discussion.

Comment on answer 5: OK

------------------------------------------------------------------------------------------------

Comment6:

Discussion should offer at least some tentative explanation for the findings. E.g. why generally OH terminations have the lowest adhesion energy on (100) surface and Obridge the highest on (111) surface.

Or why RDG has the highest adhesion energy on H terminated (111) surface while all other modeled molecules prefer O bridge on (111) surface.

What can be the reason for the order of magnitude difference between RDG (111)-H and (111)-O ontop. Similarly, RDG (100)-H and (100)-OH, NH2

Answer 6:

We have added plausible explanations for the findings in section 4. Discussion.

Comment on answer 6: OK

----------------------------------------------------------------------------------------------------------

Comment7:

Units should be given in Figure 2.

Answer 7:

We have changed “Ead” to “Eads” in Figure 2 and included “(in kJ/mol)” in the figure text for Figure 2.

Comment on answer 7: OK

----------------------------------------------------------------------------------------------------------

Comment8:

The (100) surface is favoured in two cases. This should be identified, outlined and explained.

Answer 8:

We have added an explanation to this observation in section 4. Discussion.

Comment on answer 8: OK

---------------------------------------------------------------------------------------------------------------

Comment9:

The first paragraph of the 4. Discussion section is practically the repetition of the Introduction. It is unnecessary, thus should be deleted.

Answer 9:

We have omitted this paragraph.

Comment to answer 9

There are still repetitions in the 4. Discussion which should be omitted:

Intro:

Diamond is a material with very attractive properties. These include superior electronic

properties (when doped), controllable surface termination, chemical inertness, high degree of

biocompatibility, high transparency, large electrochemical potential window, etc.”

In addition, it has been observed that, because of the intrinsic

biocompatibility and chemical stability, diamond can increase the duration of bone implants.”

A well-controlled surface termination with various species has not only proven to uphold the

cubic structure of diamond, but also to change the reactivity and properties of the diamond surface

region.”

Discussion

Diamond shows very attractive properties, e.g. biocompatibility, chemical inertness, and

controllable surface termination. It has recently been shown that diamond can increase the duration of bone implants, and has become a plausible material for bone regeneration and vascularization [6,

48,49]. The reactivity of diamond surfaces depends on various factors, such as i) diamond plane, and ii) surface termination. A controllable diamond surface termination has earlier been shown to change the reactivity and properties of the diamond surface region [43, 50]. In order to promote cell adhesion and vascularization of medical implants, the possibility to specifically modify the interaction between growth factors and the diamond surface is of largest importance.”

Please avoid these superfluous restatements because they reduce the quality of the manuscript.

Comment on new paragraph, pg. 4. section 3.1

The sentence

It adhesion energies for large physisorbed organic molecules onto differently terminated

diamond surface planes. was thereby shown that the ab initio FF method is possible to use when

calculating and comparing trends in”

should be reworded.

p { margin-bottom: 0.1in; line-height: 120%; }

Author Response

Answer 4: With groups I just meant the division of biomolecules due to their different adhesion        energies. This was just a way to simplify the rest of the presentation. It was just an artifical way to relate the various biomolecules to each other. I agree that the number of atoms in AGP1 is about 480 atoms less than for BMP2, but it is also only 420 atoms larger than fibronectin. So it is on the borderline. Since I cannot explain all details in the results, due to the complexity in surface biomolecule reactions, I hope that my version of "groups" can be accepted.

Answer 9: I have corrected the manuscript following he advice by the referee.